# Identification of a novel spinal nociceptive-motor gate control for Aδ pain stimuli in rats

Dvir Blivis[1][*][†], Gal Haspel[1,2][†], Philip Z Mannes[3], Michael J O'Donovan[1], Michael J Iadarola[4]

[1]Developmental Neurobiology Section, Laboratory of Neural Control, National Institute of Neurological Disorders and Stroke, National Institutes of Health, Bethesda, United States; [2]Federated Department of Biological Sciences, New Jersey Institute of Technology, and Rutgers, Newark, United States; [3]Molecular Recognition Section, Laboratory of Bioorganic Chemistry, National Institute of Diabetes and Digestive and Kidney Disorders, National Institutes of Health, Bethesda, United States; [4]Department of Perioperative Medicine, Clinical Center, National Institutes of Health, Bethesda, United States

**Abstract** Physiological responses to nociceptive stimuli are initiated within tens of milliseconds, but the corresponding sub-second behavioral responses have not been adequately explored in awake, unrestrained animals. A detailed understanding of these responses is crucial for progress in pain neurobiology. Here, high-speed videography during nociceptive Aδ fiber stimulation demonstrated engagement of a multi-segmental motor program coincident with, or even preceding, withdrawal of the stimulated paw. The motor program included early head orientation and adjustments of the torso and un-stimulated paws. Moreover, we observed a remarkably potent gating mechanism when the animal was standing on its hindlimbs and which was partially dependent on the endogenous opioid system. These data reveal a profound, immediate and precise integration of nociceptive inputs with ongoing motor activities leading to the initiation of complex, yet behaviorally appropriate, response patterns and the mobilization of a new type of analgesic mechanism within this early temporal nociceptive window.

*For correspondence: blivisd@ ninds.nih.gov

†These authors contributed equally to this work

Competing interests: The authors declare that no competing interests exist.

## Introduction

Assessment of responses to nociceptive stimuli in either animals or humans is a cornerstone of pain research upon which many inferences from experimental, pharmacological and clinical investigations are based (*Kruger and Light, 2010*; *Szallasi, 2010*). However, our appreciation of the full range of behavioral and physiological parameters that comprise a 'nociceptive response' remains incomplete, as does our appreciation of the integration of this information at the spinal level. This in turn affects the ability to interpret the results of interventions and manipulations of nociceptive circuits and participating molecular circuits. Time and magnitude are two, of many, domains in which human and animal behavioral responses are quantified following application of noxious stimuli and from which endpoints can be derived. However, when considering acute and chronic pain, responses in the temporal and magnitude domains occupy an extraordinarily broad range. Among the earliest, are action potentials from primary afferent nociceptive fibers and reflex withdrawal responses (*Creed and Sherrington, 1926*; *Jensen et al., 2015*; *Levinsson et al., 2002*; *Lundberg, 1979*; *Morgan, 1998*) followed by limb withdrawal reactions (*Burke et al., 1971*; *Lundberg, 1979*; *Morgan, 1998*). Over the longer-term, more integrated behavioral endpoints are measurable such as limb guarding,

**eLife digest** A bee sting or a pinprick are examples of painful experiences that trigger an immediate response in humans and other animals. Scientists have begun mapping how different parts of the nervous system control how the body reacts to pain. But there are still many questions about what happens in the very first moments after pain. For example, does the response depend on what the body is doing when the painful event occurs? Examining how animals move in response to pain may help answer these questions and possibly point to new strategies for treating pain.

Now, Blivis et al. show that the nervous system orchestrates a sequence of movements in the whole body in the first 500 milliseconds after a painful event. In the experiments, a high-speed video camera recorded what happened when rats experience a pinprick or brief burst from a hot laser on one paw. When a rat is on all four paws, it first moves it head and then picks up its foot after one of these painful experiences. In fact, the position of the rat's entire body moves to enable the head to turn towards the source of the pain. This may help the rat assess the threat and decide what to do about it.

When a rat is standing on two hind legs, however, the animal's pain reaction is delayed until the animal attains a more stable footing. The rat puts its front paws down, before moving its foot from the source of the pain. Future studies are needed to identify which parts of the brain and spinal cord are active during these early, rapid movements and if something similar happens in humans. If a similar process occurs in humans, scientists might be able to develop new pain medications that take advantage of the system that temporarily suppresses the body's immediate reaction to pain. These medications could, in future, be used to treat the heightened sensitivity to pain that can occur after an injury, or the intense "breakthrough" pain experienced by cancer patients that cannot be controlled by their usual pain medication.

changes in operant assay responding (*Neubert et al., 2005*; *Thut et al., 2007*; *Murphy et al., 2014*), modification of activity levels (*Cobos et al., 2012*; *Dolan et al., 2010*), interference with activities of daily living (*Jirkof et al., 2013*; *Rock et al., 2014*), alterations in experimental conditioned place preference (*Wagner et al., 2014*; *Xie et al., 2014*), and the production of facial grimacing (*Langford et al., 2010*; *Sotocinal et al., 2011*; *Matsumiya et al., 2012*). Additionally, the plasticity of several of these behaviorally assessed endpoints is evident when testing is performed during ongoing appetitive or attentional behaviors (*Foo and Mason, 2005*, *Foo and Mason, 2009*; *Foo et al., 2009*).

The fact that a noxious stimulus can be received anywhere on the body suggests that the noxious sensory integrator system is present at all levels of the spinal cord and trigeminal system. Ideally, nociception should be integrated with higher order neural circuits related to formulating an effective escape strategy, threat assessment and reference to previous experience and threats. Lastly, this system likely overlaps or accesses a generalized motor control circuit geared to generate rapid motor responses to abrupt environmental perturbations that might include slipping or tripping while in motion, and responses to abrupt auditory or visual stimuli and, therefore, can incorporate input from corticospinal, tectospinal, cerebello-spinal, and rubrospinal descending tracts. Appropriately integrated motor reactions to these stimuli have strong adaptive value.

Many of the tests for evaluating baseline pain, sensitization during inflammation, tissue damage, or assessing the actions of analgesic drugs or genetic manipulations, employ acute, episodic stimuli (*Mitchell et al., 2014*). Acute stimuli generally involve application of mechanical force, noxious thermal heat, or noxious cold (*Simone and Kajander, 1997*; *Cain et al., 2001*). Even here, the designation of 'acute' occupies a broad temporal range, from very brief noxious stimuli such as pin prick, an electrical pulse, or von Frey hairs (*Mitchell et al., 2010*; *Chaplan et al., 1994*), to a 100 ms laser pulse, and much longer duration tests employing progressively increasing ramp-style stimuli like the Randall-Sellito paw pressure test (*Randall et al., 1957*), hot plate responses (*Caterina et al., 1997*; *Davis et al., 2000*), or the 'Hargreaves' thermal withdrawal test (*Hargreaves et al., 1988*; *Iadarola et al., 1988*). However, the exact behavioral components that comprise, and even precede, the withdrawal endpoint have not been adequately explored. We posit that a fine-grained analysis

of the sensory-motor integrative responses to nociceptive stimuli is a critical prerequisite for defining the neural circuits that generate noci-responsive behaviors and identifying the underlying molecular, cellular, and circuit-level processes that comprise these neural programs.

In this report, we show that, in addition to the spinal segmental responses to noxious stimuli, there is a corresponding rapid and simultaneous multi-segmental, multi-limb response. Furthermore, we observe that this response can be inhibited by the animal's posture, providing evidence for a potential endogenous analgesia system that gates access of nociceptive information to the ventral horn motor neuronal withdrawal circuits. We show that this gate operates via opioid- and non-opioid-dependent mechanisms. Our observations provide several new, largely unexplored behavioral responses that can be used for assessment of acute nociceptive stimuli, effects of pharmacological treatments, modification of circuits by genetic manipulations, and how more chronic manipulations or models affect spinal excitability and the integrated sensory-motor response to noxious test stimuli. The system can be viewed as a second gate control for nociception that is permissive to the appropriate flow of information from noxious afferent input to motor output.

## Results

### Manipulations and endpoints

The results comprise three behavioral endpoints: latency to first movement, pattern of movement for the four limbs and head, and whether the stimulus caused limb withdrawal. These endpoints were measured following two stimuli: noxious heat (laser) or pinprick (sharp), and under two postural conditions: standing on all four limbs (quadrupedal stance) or when standing only on two hindlimbs (bipedal stance) (*Figure 1*). In the latter stance, we also determined the behavioral endpoints in the presence or absence of naloxone.

### Multisegmental stimulus-evoked behavioral responses for heat and mechanical stimuli

#### Quadrupedal stance

The movements of five body parts in response to laser and sharp stimulation of the fore and hindlimbs are analyzed in *Figure 2*. In panel A, the overall movement patterns for each limb and the head are depicted; in Panel B, both the latency of the first movement of each limb and, for the stimulated limb, the elaboration of a twitch or a full hind paw/limb withdrawal are quantified (*Figure 2C*). The behavioral responses to the laser and pinprick stimuli (panel A) were quantified by measuring the latency either from the end of the thermal pulse stimulus or from the moment of contact with the sharp, pin prick stimulus to the onset of the limb and head movement. The duration of the movement is also shown. Each line on the graphs in *Figure 2A* is color-coded for the respective body part. Upon forelimb stimulation, in the quadrupedal stance, the rat produced an appropriate withdrawal sequence in 100% of the trials for both laser (n = 5) and sharp (n = 14) stimuli (*Figure 2A,B*; see *Videos 1,2*). Eighty percent (laser) to 100% (pinprick) of forepaw stimulations provoked orientation of the head toward the stimulated limb. Despite movement of the head, forelimb withdrawal occurred without movement of either hindlimb. This was especially evident with the thermal laser pulse: neither hindlimb moved following the stimulus (*Figure 2A,Ba*). The sharp pin prick stimulus triggered small movements, mainly of the toes, in 30 to 50% of the trials in the contralateral fore and hindlimbs (*Figure 2Bc*, green symbols). Also evident is a difference in the latency of the first movement or limb withdrawal between sharp and thermal laser stimulation. Movements evoked by the sharp stimulus always occurred earlier than those evoked by the thermal stimulus by 48.3 ms and 48.1 ms for the stimulated forepaw or the head, respectively.

The most complete response pattern occurred when a hind paw was stimulated (see *Video 3* and *Video 4*). Hind paw thermal laser (n = 29) or sharp (n = 31) stimulation generally involved all limbs and the head in a consistent temporal sequence. With both stimulus modalities, the head and the stimulated hind paw moved at approximately same time (Pinprick: Head- $52.0 \pm 5.2$ ms; ipsi Hind paw-$51.6 \pm 3.6$ ms; Thermal Laser: Head- $169.9 \pm 9.5$ ms; ipsi Hind paw- $180.2 \pm 8.1$ ms). As was observed for forepaw stimulation, there was a noticeable difference in the withdrawal latency for the two stimuli: Sharp stimulation provoked a withdrawal movement within ~50 ms, whereas thermal laser stimulation caused withdrawal within ~180 ms (*Figure 2B*). These differences in timing are

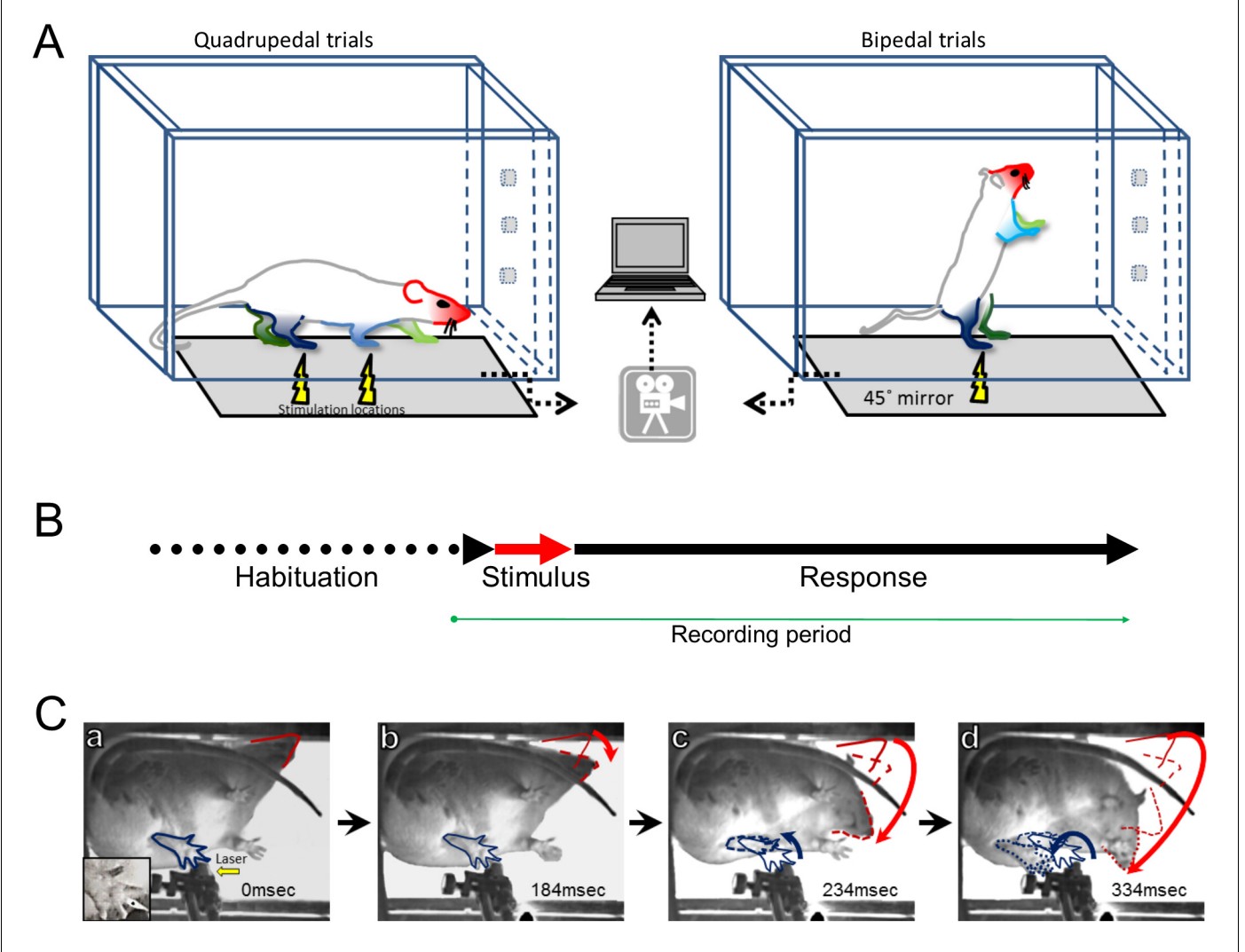

**Figure 1.** Experimental design and protocol. (A) Adult unrestrained rats were placed in a Plexiglas chamber which allows them to move freely. The chamber was placed on an elevated glass or a wire mesh surface depending on the stimulus type (i.e. laser; sharp or blunt). Stimulus location is indicated by the yellow lightning symbol and the limbs and head are color coded. Selective thermal activation was achieved using a brief (100 ms), small-diameter (1.6 mm), and high-energy (5000 mA) infrared diode laser pulse. Pinprick stimuli were applied by pushing a sharp pin against the foot pad until the skin was dimpled but not penetrated and a rounded wooden stick was used for non-noxious blunt stimulus. A high-speed camera captured the animal's postural changes via a mirror under the platform surface. The collected data were transferred and stored on a computer for off-line analysis. (B) The experimental protocol had three phases: habituation– allowing the rat to get used to the chamber and the required posture (on four or two limbs); stimulus – sensory stimulation was delivered to the target limb; response– a 500–800 ms window where any postural changes or movements were documented. (C) A typical response to a thermal Aδ afferent activation. A sequence of video frames (a–d) demonstrating the response to a laser pulse applied to the left hind limb of rat standing on four limbs (*Video 1*) . For simplicity, only the stimulated limb (blue) and the head (red) are marked. Immediately after the stimulus (a– 0 ms), all limbs are in place and no movement has occurred. Bottom left insert: zoom in to the stimulated hindpaw where the black dot represents the laser target. The head was the first body part to move (b), followed by the movement of the stimulated limb (c) while the head continued moving toward the stimulated area. Red and blue arrows demonstrate the overall trajectory of the head and left hindlimb movement, respectively.

evident in both the limb movement histograms and in the mean first response scatter plots (*Figure 2A and Bb, Bd*). Thus, processing of the stimuli into an integrated response is much more rapid with a mechanical stimulus than with a thermal stimulus, even with the very fast rate of stimulation provided by the infrared diode laser.

## Transition to withdrawal

We observed that an immediate withdrawal did not always occur upon application of the noxious stimuli especially when delivered to the hindlimb (see *Table 1*). Thus, in addition to limb movement patterns and latencies, variation in a third parameter, referred to as the transition to the withdrawal response was measured (*Figure 2C*). In the quadrupedal stance, laser and sharp stimulation of the forepaw produced a brisk limb withdrawal in 100% of the trials (Laser, n = 5; Sharp, n = 14) without any preceding twitch of the toes or paw. Sharp stimulation of the hindlimb also produced a 100% withdrawal and was preceded by twitch in 7% of the trials (n = 31). In contrast, infrared thermal stimulation of the hind paw failed to produce withdrawal in 18% of the trials, with 14% showing just a twitch and 4% no movement at all (*Figure 2C*). Thus, in most of these 'failure trials', evidence for reception of the stimulus and its processing to cause a motor response (i.e. an evoked twitch) is observable, suggesting the stimulus did not reach threshold for a full withdrawal or that it was actively inhibited. We do not think that the twitches are a reflex response to proprioceptive sensory fibers activation since we detect twitches in the hindlimbs following a laser stimulus that does move the digits, and we do not observe twitches when a sharp stimulus is applied to the forelimbs.

The presence of this suppressive circuit becomes more evident when the animal is in the bipedal or reared-up stance and the results from these trials are analyzed in more detail in the Bipedal Stance section.

## Latencies of early movements

To evaluate statistically the differences between the movements evoked by stimulation of the fore- and hind paws, we compared the latency of the earliest movement of the head or a limb to the various stimuli. *Figure 3* and *Table 2* show that, for a sharp stimulus, while the withdrawal latency for hind paw stimulation was on average longer than for forelimb stimulation, the difference was not statistically significant (forelimb: 33.2 ± 13.0 ms, n = 14; hindlimb: 46.9 ± 21.0 ms, n = 31; p>0.5, ANOVA, Tukey's multiple comparison test). By comparison, laser heat stimuli that preferentially excites Aδ fiber afferents produced different response latencies for the fore and hindlimbs: the latency to first movement upon stimulation of the hindlimb was 167.5 ± 43.0 ms (n = 29) compared to stimulation of the forelimb (93.9 ± 33.0 ms, n = 5; p<0.0001).

## Head movement

*Figure 2* reveals the consistent and early occurrence of head movements as a component of the nociceptive response. Head movement occurred nearly simultaneously and could even precede movement of the paw receiving the noxious stimulus. Early head movement occurred with either fore paw or hind paw stimulation and with any type of stimulus (sharp, laser, or blunt; *Figures 2* and *4*). The mean latency to head movement upon sharp stimulation of the forelimb was 40.3 ± 7.0 ms and for hindlimb stimulation was 52.8 ± 5.0 ms. The mean latency to the start of head movement upon laser heat stimulation of the forelimb was 88.5 ± 17.0 ms and 169.9 ± 10.0 ms for the hindlimb stimulation. The head movement precedes movement of the stimulated limb for heat delivered to either the fore paw or the hind paw (fore paw- 95.4 ± 14.0 ms; hind paw- 180.2 ± 8.0 ms), although the time difference was not significant. These data are consistent with the idea that the motor apparatus governing directed head movement is temporally integrated with limb withdrawal responses to abrupt noxious stimuli.

## Non-noxious blunt mechanical

As a negative control, the multisegmental responses to a non-noxious tactile stimulus were also examined (*Videos 5* and *6*). The blunt stimulus was a wooden stick with a rounded conical tip that was not sharp and did not have distinct edges like those on von Frey hair filaments. The rats were largely indifferent to this non-noxious stimulus. When they did remove their paw, it occurred after the stick elevated the paw above the wire mesh platform and it was without the alacrity that was characteristic of the noxious laser heat or pinprick (*Figure 4*). Nonetheless, the first movement (head orientation) occurred within 226.6 ± 77.0 ms and 263.6 ± 83.0 ms of pushing on the fore- or hind paw, respectively. The indifference to the blunt probe is evidenced by the fact that in 50% of the blunt trials the animals terminated the trial simply by walking away. The limb movement latencies were significantly longer than the responses to laser or the pinprick stimulus (blunt-forelimb:

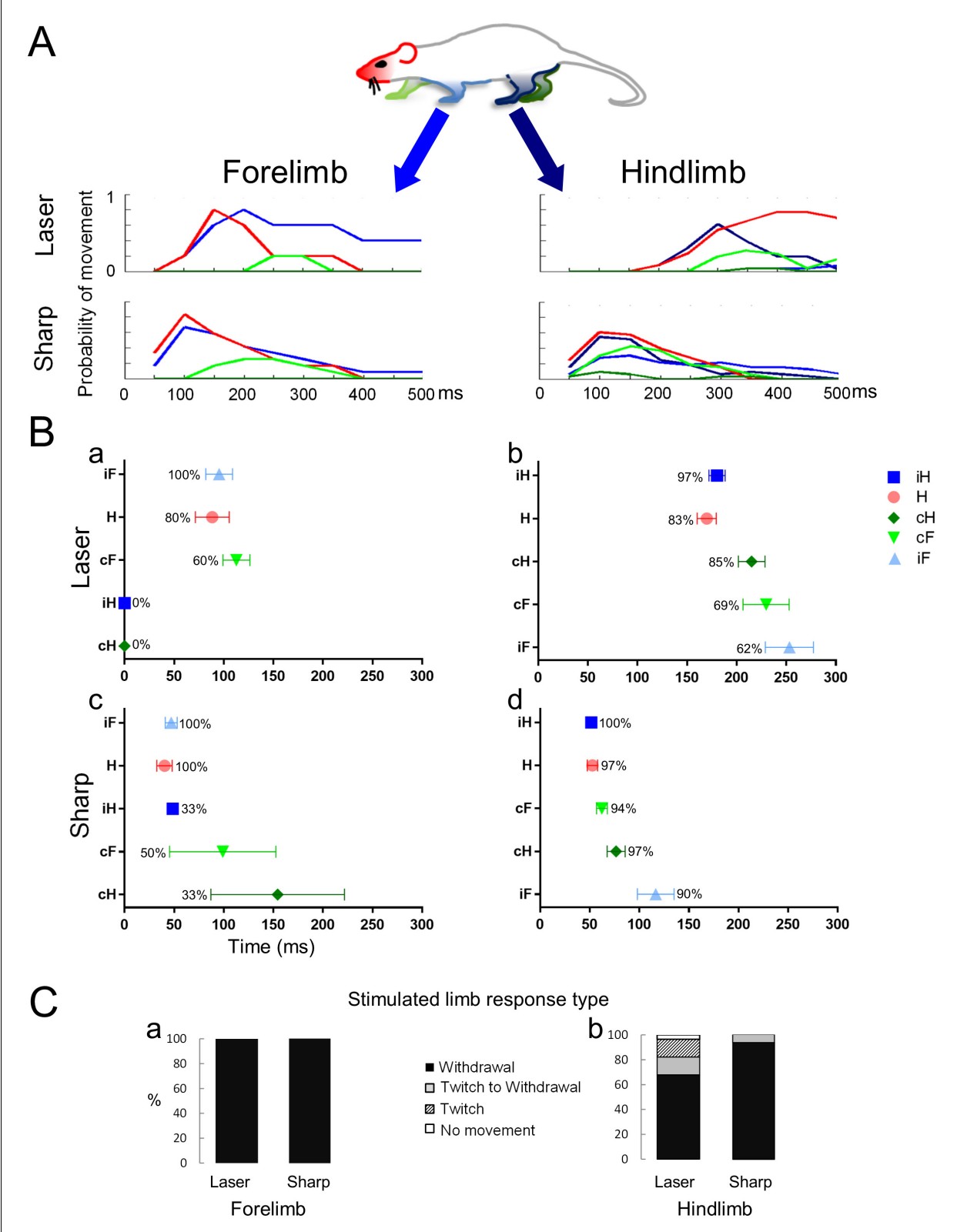

**Figure 2.** Noxious stimuli evoke a complex multi-segmental response. The graphs show the temporal progression of the head and limb responses to a noxious stimulus applied to the forelimb or to the hindlimb. (**A**) Color codes for the head and the limbs and movement histograms showing the probability of a head or limb movement in 50 ms bins. (**B**) Scatter plots showing the mean first response (± S.E.M) of the movements and the probability of a movement (%) evoked by stimuli given to the forelimb (Ba and Bc) or the hindlimb (Bb and Bd). The averaged first response time for the stimulated

*Figure 2 continued on next page*

*Figure 2 continued*

limb is on top (ipsilateral forelimb for Ba and Bc; ipsilateral hindlimb for Bb and Bd). The other limbs and the head are presented according to their appearance after the stimulus. Notice that the head is the most likely to move regardless the stimulus type or location and that the hindlimbs do not move when the laser stimulus is applied to a forelimb paw. (C) Bars representing the frequency distribution of the response type (full withdrawal, twitch to withdrawal, twitch only, and no movement of the stimulated limb) of the stimulated limb. The stimulated forelimb responded with a full withdrawal 100% of the times to both sharp and laser stimuli. H– Head (●); iF- ipsilateral forelimb (▲); iH– ipsilateral hindlimb (■); cF– contralateral forelimb (▼); cH– contralateral hindlimb (♦).

308.1 ± 241.0 ms, n = 7; blunt- hindlimb: 324.1 ± 115.0 ms, n = 6) and the probability of withdrawal was substantially lower (blunt stimulus to forelimb: 57%; blunt stimulus to hindlimb: 33%; no difference in the probability to respond: forelimb vs hindlimb p=0.6; Fisher Exact test; 2 × 2 contingency table for two nominal variables with expected small or zero values).

## Gating of nociceptive withdrawal response by posture
### Bipedal stance

In the next set of experiments, we asked whether the animal's response to a noxious stimulus was affected by its posture at the time of stimulation. We used the same experimental protocol for sharp and laser stimuli, except that the hind paws were stimulated only when the animal was in a bipedal stance (i.e. standing on its hindlimbs; *Figure 1A*; *Video 7* and *Video 8*). Under these conditions, as seen in the movement histograms, the multisegmental response was clearly modified and frequently blocked completely (*Figure 5A,B,C*). When the animals were standing on all four limbs, they responded to a laser stimulus in 83% of the trials (n = 29) and to sharp stimulus 100% of the trials (n = 31, sharp vs laser, p<0.05). However, when the animals stood only on their hindlimbs, they responded to a laser stimulus in only 49% of the trials (n = 39) (p<0.01; *Figure 5A*). We also found a statistical difference in the proportion of trails in which the animals responded to the sharp stimulus in the quadrupedal (n = 31) compared to the bipedal stance (n = 45) (100% vs. 67%, p<0.001; *Figure 5A*). The latency to the onset of head or limb movements when the sharp stimulus was applied, was significantly longer in the bipedal compared to the quadrupedal stance (Sharp stimulus-standing on hindlimbs vs. standing on all four limbs: 86.3 ± 11.0 ms vs 46.9 ± 4.0 ms, respectively, p<0.005, Two-ways ANOVA, Tukey's multiple comparison test) (*Figure 6B*).

What is not evident from the latency measurements is how, or the process by which, the withdrawal of the hind paw occurs when the animal is standing on both hindlimbs. An integrated, orchestrated set of motor events occurred within a window of 50 to 80 ms starting from the first response to the sharp pinprick or the laser stimulus, respectively. When the rat was in a bipedal stance, movements of the stimulated hindlimb (twitching, withdrawal, shaking) were delayed until the forepaws came down and touched the platform. In some cases, movement occurred while the forelimbs were returning to the platform. When in the bipedal stance, the animals reacted significantly faster to sharp stimuli than to laser stimuli delivered to the hind paw (sharp vs. laser 86.3 ± 11.0 ms, n = 45 versus 150.3 ± 11.0 ms, n = 39; p≤0.0001, two-way ANOVA, Tukey's multiple comparison test;

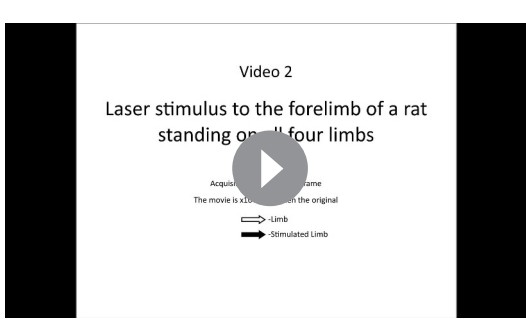

**Video 1.** Forelimb laser stimulus when the rat is standing on all four limbs.

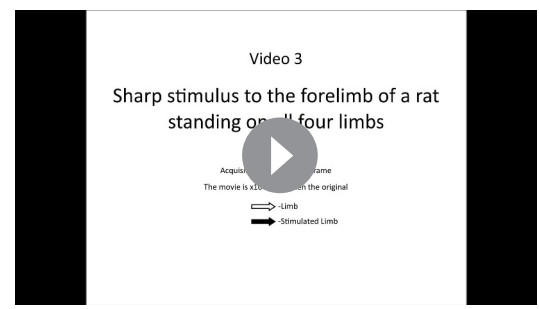

**Video 2.** Forelimb sharp stimulus when the rat is standing on all four limbs.

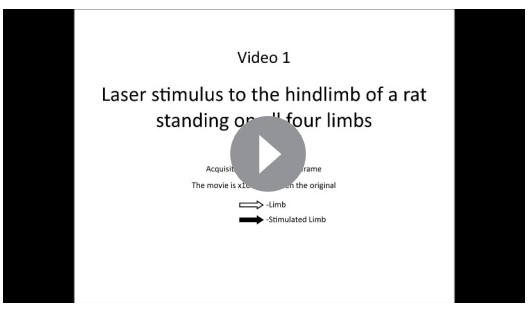

**Video 3.** Hindlimb laser stimulus when the rat is standing on all four limbs.

**Video 4.** Hindlimb sharp stimulus when the rat is standing on all four limbs.

*Figure 6B*). These results are consistent with the temporal differences between sharp and thermal laser stimulation seen in the quadrupedal stance.

## Effects of bipedal stance on the type of withdrawal movement and latency

The bipedal posture suppresses or delays the withdrawal response to the two noxious stimuli until the animal shifts from a bipedal stance to a quadrupedal stance. The withdrawal reaction, when it

**Table 1.** Comparison between the response rate for different sensory stimuli applied to the fore– or hindlimb of a rat standing on four or two limbs. Results of Fisher Exact test, 2 × 2 contingency tables. Values in red are significant. Values in black are not significant. In parenthesis is the rate for full withdrawal for each condition.

| | | | Laser | | | | Sharp | | | |
| --- | --- | --- | --- | --- | --- | --- | --- | --- | --- | --- |
| | | | Standing on four | | Standing on two | | Standing on four | | Standing on two | |
| | | | Forelimb (%100) | Hindlimb (%82.8) | Hindlimb (%48.7) | Hindlimb +NL (%63.2) | Forelimb (%100) | Hindlimb (%100) | Hindlimb (%66.7) | Hindlimb +NL (%55) |
| Laser | Standing on four | Forelimb (%100) | | | | | | | | |
| | | Hindlimb (%82.8) | 1.0 | | | | | | | |
| | Standing on two | Hindlimb (%48.7) | >0.05 | <0.01 | | | | | | |
| | | Hindlimb +NL (%63.2) | >0.2 | >0.1 | >0.4 | | | | | |
| Sharp | Standing on four | Forelimb (%100) | 1.0 | >0.1 | <0.001 | <0.05 | | | | |
| | | Hindlimb (%100) | 1.0 | <0.05 | <0.001 | <0.001 | 1.0 | | | |
| | Standing on two | Hindlimb (%66.7) | >0.3 | >0.1 | >0.1 | >0.7 | <0.05 | <0.001 | | |
| | | Hindlimb +NL (%55) | >0.1 | >0.5 | >0.7 | >0.7 | <0.001 | <0.001 | >0.4 | |

**Source data 1.** Proportion of withdrawals for different sensory stimuli applied to the fore- or hindlimb of a rat standing on four or two limbs. Source data and Column histogram presenting the number of trials the rats responded with a full withdrawal to a stimulus. Each column presents the total number of trials. (Black: full withdrawal; Gray: no withdrawal). l-laser stimulus; s-sharp stimulus; 4-animal on all four limbs; 2-animal on two hind limbs; f -forelimb stimulation; h -hindlimb stimulation.

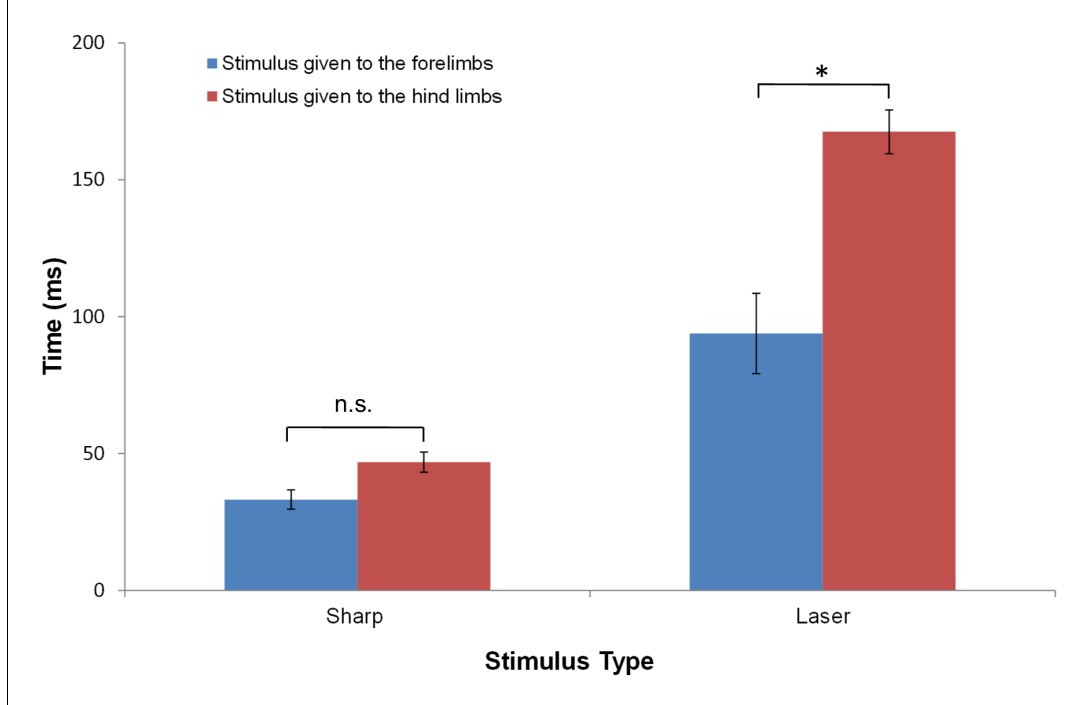

**Figure 3.** The average response latency to Aδ noxious stimulus is dependent on the stimulus modality and location. The bar plots show the average latency and standard error of the mean for the first response of any limb following a sharp or a laser stimulus to the fore- (blue) or hind- (red) limb of a rat standing on all four limbs. Collectively, the data show that the earliest responses were to sharp stimuli and the slowest were to laser stimuli. The averaged first response latencies for forelimb stimulation were shorter than those for hindlimb stimulation. The shortest latency was found when the sharp stimulus was applied to the forelimb (33.2 ± 13.0 ms, n = 14) or to the hindlimbs (46.9 ± 21.0 ms, n = 31). These latencies were not significantly different. A significant difference in the latencies between forelimb and hindlimb stimulation was found for the responses to laser stimulation (Forelimb: 93.9 ± 33.0 ms, n = 5; Hindlimb: 167.5 ± 43.0 ms, n = 29; *p<0.0001). (Two-way ANOVA; Tukey's multiple comparisons test).

does occur, is inhibited until the forepaws are placed on the test platform or are about to contact the platform. This is evident in the delay to hindlimb withdrawal seen with the sharp stimulus where withdrawal occurs after ~50 ms in the quadrupedal stance, whereas, when the animal is standing on its hindlimbs, the withdrawal of the stimulated hindlimb occurs after ~90 ms. The additional 40 ms allows the animal's posture to be stabilized (i.e. all four limbs are in contact with the platform prior to withdrawal).

Further analysis of the trials, revealed three types or levels of withdrawal inhibition when in the bipedal stance: complete, delayed, and partial (see bar plots in *Figure 5Cb and Cd*). Complete inhibition means that no movement of the stimulated limb was seen after nociceptive stimulation. Delayed inhibition was characterized by an initial twitch followed by full withdrawal of the limb. Partial inhibition was characterized by a twitch, usually of a toe on the stimulated paw, but no withdrawal of the limb. At least one of the three types of inhibition occurred in ~50% of all trials administered to rats in the bipedal stance for both thermal laser and sharp stimuli.

### Effects of naloxone on behavioral responses to noxious stimuli in the bipedal stance

We hypothesized that the postural inhibition and slowing of the behavioral responses to noxious stimuli might result from modulation of an endogenous opioid peptide-mediated suppression of the withdrawal reflex and orienting movements. To test this idea, we injected animals systemically with naloxone, an opioid receptor antagonist (*Videos 9* and *10*). Contrary to our expectation, the response rate to noxious stimuli was unchanged in the presence of naloxone (laser: 63%, n = 19; sharp 55%, n = 20, *Figure 6A*) when the rat was standing on its hindlimbs, compared to pre-drug control conditions (laser no drug vs. naloxone, p=0.4026; sharp no drug vs. naloxone, p>0.4; laser naloxone vs. sharp naloxone, p>0.7). The presence of naloxone also did not change the average

withdrawal latency to thermal laser stimulation when comparing three conditions (quadrupedal, bipedal, and bipedal + naloxone) (*Figure 6B*). However, naloxone significantly reduced the latency to the first movement in response to the sharp stimulus (no drug, quadrupedal: 46.9 ± 4.0 ms; no drug bipedal: 86.3 ± 11.0 ms; bipedal with naloxone: 55.1 ± 7.0 ms, p<0.05; *Figure 6B*).

One limitation of these studies is that we did not use a vehicle injection as a control for the naloxone injection. Injections are anxiogenic and could influence the timing of withdrawal responses to noxious stimuli (*Lapin, 1995*). We note, however, that the latency of the first response to a thermal stimulus was unchanged following naloxone injection, suggesting that the effects of the injection per se did not alter the responses to all noxious stimuli. Despite this observation, we cannot eliminate the possibility that the injection might have influenced the responses to the pin prick stimulus and we consider this potential confound in more detail in the discussion.

## Discussion

The paw withdrawal reflex and the escape response have been used extensively as a behavioral assay for pain in animal models. The spinal mechanisms underlying these early and rapid responses to noxious stimuli have been studied in detail in anesthetized or sedated animals or various decerebrate or spinal preparations (*Lundberg, 1979*; *Shin et al., 1986*; *Granmo et al., 2013*; *Kelly et al., 2013*). Much less is understood, however, about when and how these early reflexes are integrated into the more complex behavioral reactions to a noxious stimulus in the unrestrained conscious animal. In this paper, we identify a multi-segmental response to thermal laser and sharp stimuli. Both types of stimuli activate Aδ fibers (*Mitchell et al., 2010*) and trigger withdrawal of the stimulated limb coupled with movements of the other limbs appropriate for maintaining the animal's posture. Furthermore, we describe an inhibitory mechanism that temporarily suppresses segmental nociceptive responses until an appropriate quadrupedal stance is achieved. Obviously, such interactive responses cannot be observed in anesthetized animals. Thus, the complexity of the 'reflex withdrawal response' generally has been underestimated, leading to a constrained frame of reference for acute nociceptive studies. Our data indicate the need to extend studies of nociception beyond the familiar segmental framework and, in future studies, to investigate the neural circuitry responsible for the multisegmental responses and how they are integrated with inputs from higher centers (*Schouenborg et al., 1995*; *Stepien et al., 2010*; *Arber, 2012*; *Levine et al., 2014*; *Pivetta et al., 2014*; *Hilde et al., 2016*).

### Nociceptive responses while in quadrupedal stance

#### Initial movements

We first consider the stimuli delivered to the animal standing on all four limbs. Under these conditions, the earliest movement in response to a noxious stimulus was of the head toward the stimulation site. This was followed, or nearly simultaneously accompanied by, movement of the stimulated

**Table 2.** Comparison between the average first response latencies to different sensory stimuli applied to the fore- or hindlimb of a rat standing on four limbs. Results of two-way ANOVA analysis with Tukey's multiple comparisons test. In parenthesis is the average latency and standard deviation per each condition.

|  | Sharp–Forelimb (33.2 ± 13.0 ms) | Laser—Hindlimb (167.5 ± 43.0 ms) | Laser—Forelimb (93.9 ± 33.0 ms) |
|---|---|---|---|
| Sharp– Hindlimb (46.9 ± 21.0 ms) | ns | <0.0001 | <0.05 |
| Sharp–Forelimb (33.2 ± 13.0 ms) |  | <0.0001 | <0.01 |
| Laser– Hindlimb (167.5 ± 43.0 ms) |  |  | <0.0001 |

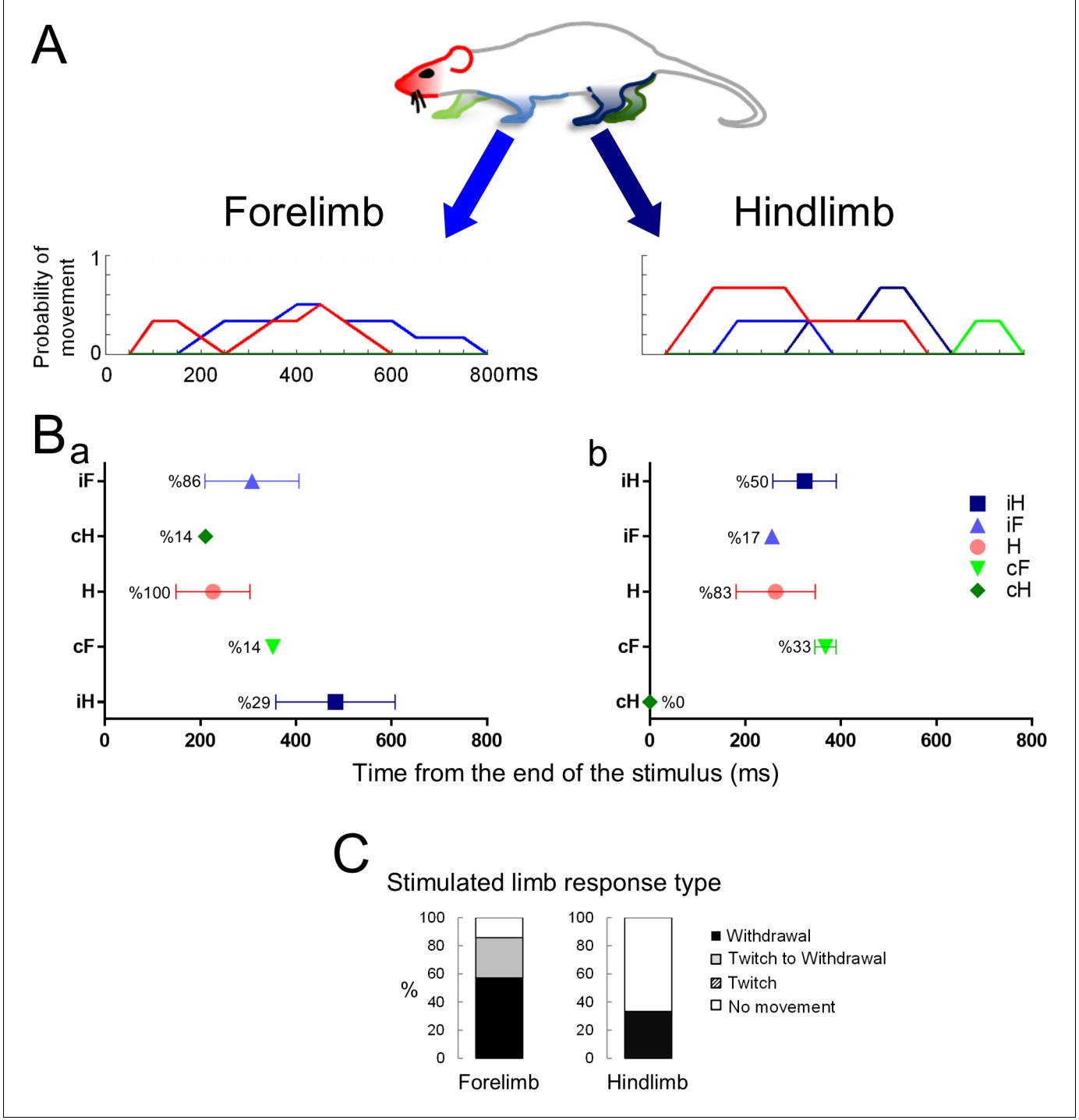

**Figure 4.** The response to a blunt stimulus when the animal is standing on four legs. Movement histograms (**A**) and averaged first response scatter plots with the probability (%) of a movement (**B**) showing the movements evoked by stimuli to the forelimb (left panels) or hindlimb (right panels) paws. (**C**) Bar plots representing the proportion of withdrawals (■), twitches (■), or no movements (□) of the stimulated limb. Color codes for the head and the limbs as in *Figure 2*. H– Head; iF- ipsilateral forelimb; iH– ipsilateral hindlimb; cF– contralateral forelimb; cH– contralateral hindlimb.

limb and the contralateral forelimb and then by movement of the contralateral hindlimb and ipsilateral forelimb. When a quadrupedal animal raises one limb it assumes a diagonal posture in which the weight is borne primarily by the contralateral hindlimb and the diagonal (ipsilateral) forelimb. This behavior was initially reported in dogs (*Ioffe and Andreev, 1969*); cited in *Gahéry and Massion*

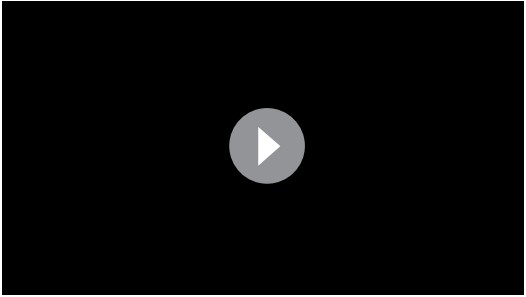

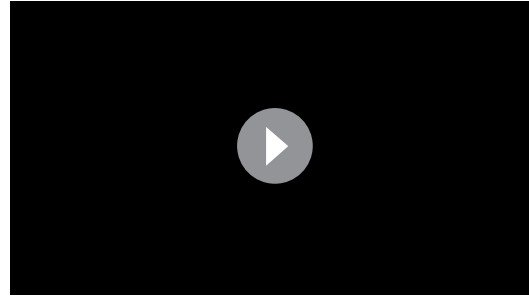

**Video 5.** Forelimb blunt stimulus when the rat is standing on all four limbs.

**Video 6.** Hindlimb blunt stimulus when the rat is standing on all four limbs.

*(1981)* and later confirmed in cats (*Coulmance et al., 1979*; *Massion and Gahery, 1979*; *Gahéry and Massion, 1981*). Consistent with these reports, when a noxious stimulus was applied to the hind paw, the sequence of movements elicited resembles a diagonal postural adjustment. Thus, the pattern of earliest limb movements involved the stimulated hindlimb and its diagonal, contralateral forelimb, with the remaining limbs supporting the animal's weight. Eventually, the other diagonally opposed limbs also moved. This sequence of movements was similar for both pin prick and laser stimulation except that the pinprick responses were of shorter latency and duration compared to the responses of the laser stimulus. A significant latency difference in the withdrawal latency for thermal laser stimulation was found between stimuli applied to the fore paw versus the hind paw (*Figure 3*). Multiple factors may account for this difference including the physical length of the nerve (*Mitchell et al., 2010*), the higher innervation density of myelinated fibers in the fore paw compared to the hind paw (*Matsumoto and Mori, 1975*) and the wide range of conduction velocities for Aδ fibers (*Ringkamp and Meyer, 2008*; *Abraira and Ginty, 2013*). Although the withdrawal latency for the sharp stimulus was slightly longer for the hind paw, the difference was not significant when assessed with ANOVA.

## Subsequent movements

It is worth noting that the subsequent movements of the hind and fore limbs diagonal to the stimulated hindlimb are also consistent with the animal initiating a locomotor step to avoid the stimulus. Such a response has been observed before in Sherrington's studies of decerebrate and spinalized cats (*Sherrington, 1910*).

The response to noxious stimulation of the forelimbs was less extensive than the responses to stimulation of the hindlimbs. We observed movement of only the head and forelimbs. This is consistent with early studies of long reflexes by Sherrington who showed that forelimb stimulation was much less likely to initiate movement of the hindlimbs than vice versa (*Sherrington, 1910*).

## Bipedal stance
### Inhibition of withdrawal

We found that the behavioral responses to noxious stimuli were dependent on the animal's posture when the stimulus was applied. Fewer animals reacted to the noxious stimuli when standing on hindlimbs, and those that did, produced fewer limb movements than those evoked in the quadrupedal posture. This was particularly evident for the sharp stimulus. Movement of the head and hindlimbs was delayed or suppressed completely compared to the quadrupedal stance. The simplest interpretation of these findings is that, while in the upright stance, the animal temporarily inhibited the limb responses to maintain postural stability. In the cases when withdrawal did occur while in the upright stance, further review of the video recordings disclosed stabilization via placement of the forelimbs on the sidewall of the test enclosure and in these cases the latencies of the head and limb movements were similar to the quadrupedal latencies.

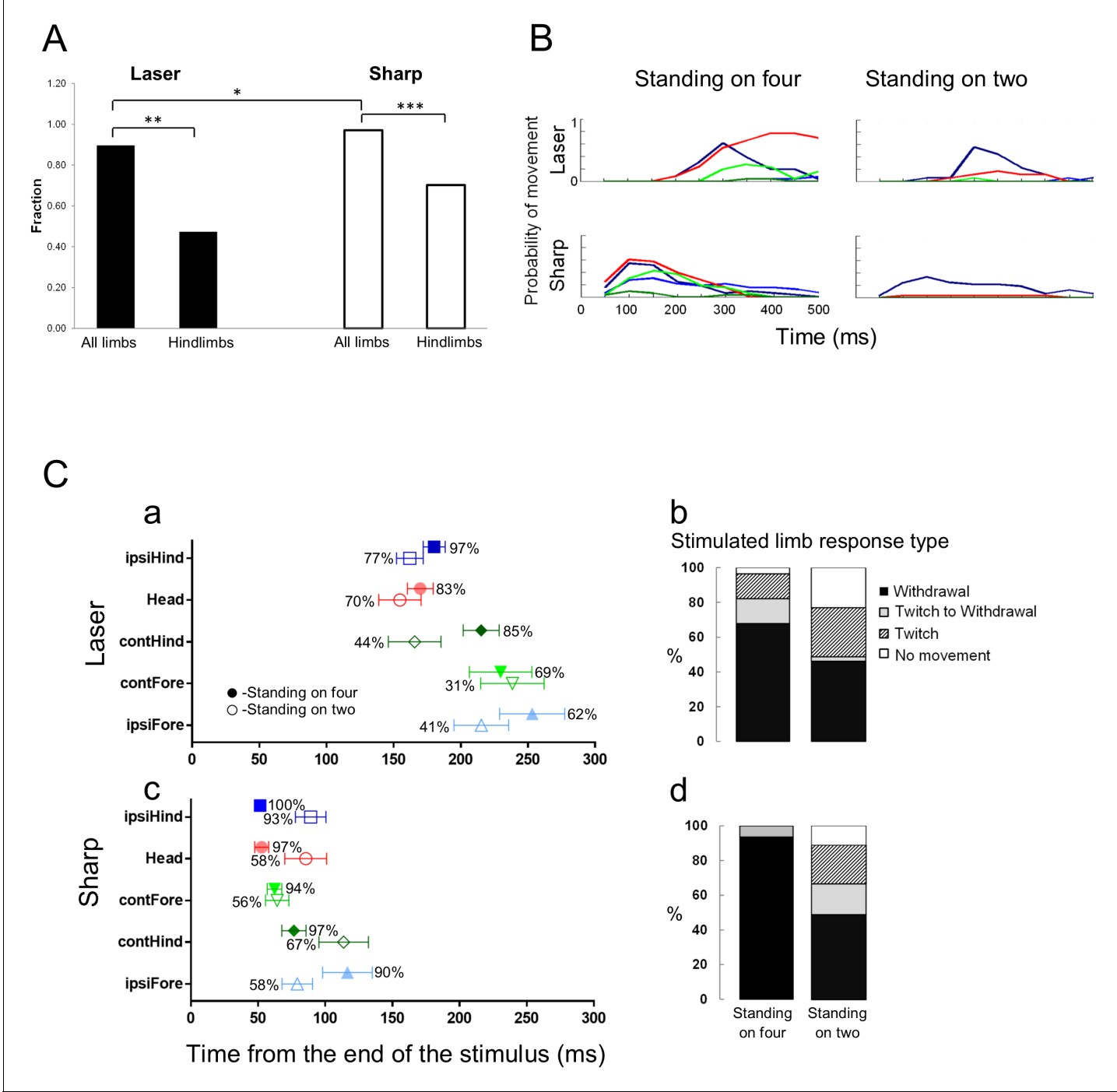

**Figure 5.** Standing on hindlimbs decreased the response frequency to noxious stimuli and resulted in fewer stimulus-evoked limb movements. Comparison of the response rate to laser or sharp stimulus given to the hindlimbs under two different postural conditions. (**A**) A significant difference in the response rate was found when the rats were standing only on their hindlimbs compared to all four limbs. The proportion of rats responding to the laser or sharp stimulus was significantly lower when the rats were standing on their hindlimbs (laser on four vs. laser on hindlimbs, \*\*p<0.01; sharp on four vs. sharp on hindlimbs, \*\*\*p<0.001; Fisher Exact test 2 × 2 contingency table, see also **Table 1**). A significant difference was also found between the response rate to laser and sharp stimuli when the rats were standing on all four limbs (\*p<0.05). No significant difference was found between the fraction of rats responding to laser stimulus compared to the sharp stimulus (laser on hindlimbs vs. sharp on hindlimbs, p>0.1). (**B**) Movement histograms show a reduction in the number of moving limbs when the rat is standing on its hindlimbs (the probability histograms on the left are adapted from **Figure 2a** for comparison). (**C**) The effects of posture on the averaged first response (plots with probability of movement (%)) (Ca, Cc) and the stimulated limb response type distribution (Cb and Cd). Filled symbols represent the average first response when the rat is in quadrupedal

*Figure 5 continued on next page*

*Figure 5 continued*
stance (from *Figure 2* for comparison). Unfilled symbols represent the average first response when the rat is rearing on its hindlimbs. (Data presented on the left part of B and C is adapted from *Figure 2* for comparison).

## Effects of naloxone

We hypothesized that if an endogenous opioidergic network participated in the postural inhibition, then naloxone would reverse or attenuate the inhibition. However, in animals pretreated with naloxone we did not observe a reversal of the posture-dependent inhibition of paw withdrawal. The multi-segmental responses were still suppressed and we did not detect a facilitation of the response rate for withdrawal. However, examination of the first movements following the sharp stimulus revealed a significant naloxone-dependent decrease in latency. This result suggests that the increase latency to withdrawal in the bipedal stance may be due to, at least in part, to direct release of opioid peptides. As we discussed earlier, we did not use saline-injections as a control for the naloxone injections. The effect of naloxone on the latency to the first movement was only observed for the pin prick stimulus, suggesting that the injections themselves did not have non-specific analgesic effects. This may be because the experimental testing was done 30 min after the naloxone injection, which exceeds duration of analgesia evoked by many mild stressors or by intrathecally administered opioid peptides such as enkephalin or other proenkephalin-derived peptides (*Iadarola et al., 1986*). Consistent with this conclusion, *Bryant et al. (1983)* found that intrathecal saline injections did not alter the reaction latency to hot-plate test or the threshold in a paw-pressure test. While we cannot exclude a non-specific effect of the injection, our data favor the existence of an opioidergic component to the postural inhibition of the withdrawal reflex.

An opioidergic component to the postural inhibition is consistent with studies showing that withdrawal reflexes are under tonic inhibition by opioids which can be relieved by naloxone (*Catley et al., 1983*; *Chung et al., 1983*; *Clarke et al., 1992*; *Steffens and Schomburg, 2011*). In the present test system, however, naloxone does not alter all aspects of inhibitory postural gating. Opioids interact with multiple non-nociceptive pathways (*Steffens and Schomburg, 2011*), and can modulate locomotor-like activity in the neonatal rat spinal cord (*Blivis et al., 2007*), and met5-enkephalin-Arg6-Gly7-Leu8, a proenkephalin-derived met5-enkephalin-Arg6-Gly7-Leu8 containing immunoreactive terminals can be detected on the perikarya of motor neurons (*Iadarola et al., 1985*, *1988*). Furthermore, noxious stimuli (*Blivis et al., 2007*) or ventral root stimuli (D Blivis; unpublished data) that initiate locomotor-like activity in the isolated spinal cord of the neonatal rat and mouse are blocked by the μ-opioid agonist DAMGO. Accordingly, we hypothesize that the latency of the first movement is regulated by an opioid action at the spinal level that may include deep dorsal horn laminae and perhaps also the ventral cord. To account for the complete postural inhibition of the withdrawal response, we propose that other glycinergic and GABAergic inhibitory networks are engaged (*Levine et al., 2014*; *Harvey et al., 2004*; *Foster et al., 2015*). The fact that glycine and GABA antagonists are convulsant agents and produce nocifensive behaviors when administered intrathecally precludes direct pharmacological investigation of these two candidate transmitters.

It is notable that we did not see an effect of naloxone on the latency to first movement with laser stimulation. We attribute this to several factors. One is timing of the afferent volley into the spinal cord. With the sharp stimulus, afferent activation is coincident with the pinprick and the timing is temporally delimited. Laser stimulation requires a minimum of ~60 ms of heating before reaching a temperature that causes paw withdrawal (*Mitchell et al., 2010*, *2014*). This comparatively long duration may contribute to an asynchronous activation of the thermo-nociceptors, whereas the pinprick yields an immediate, synchronous volley. The mechanically responsive Aδ fibers have fast conduction velocities (*Ringkamp and Meyer, 2008*) which will also contribute to a temporally discrete activation and which could facilitate our ability to detect a change in latency to first movement. Consistent with the idea of distinct fiber types, at the molecular level, the composition of receptors and ion channels may also be different (*Goswami et al., 2014*; *Usoskin et al., 2015*). Further studies on the anatomical connectivity of the two afferent fiber populations can clarify their potential for direct or polysynaptic connectivity to spinal premotor neurons at the segmental level. Although the preponderance of neuro-anatomical data indicates that the connections between the superficial laminae to the deeper laminae is not direct (*Cui et al., 2016*; *Hilde et al., 2016*).

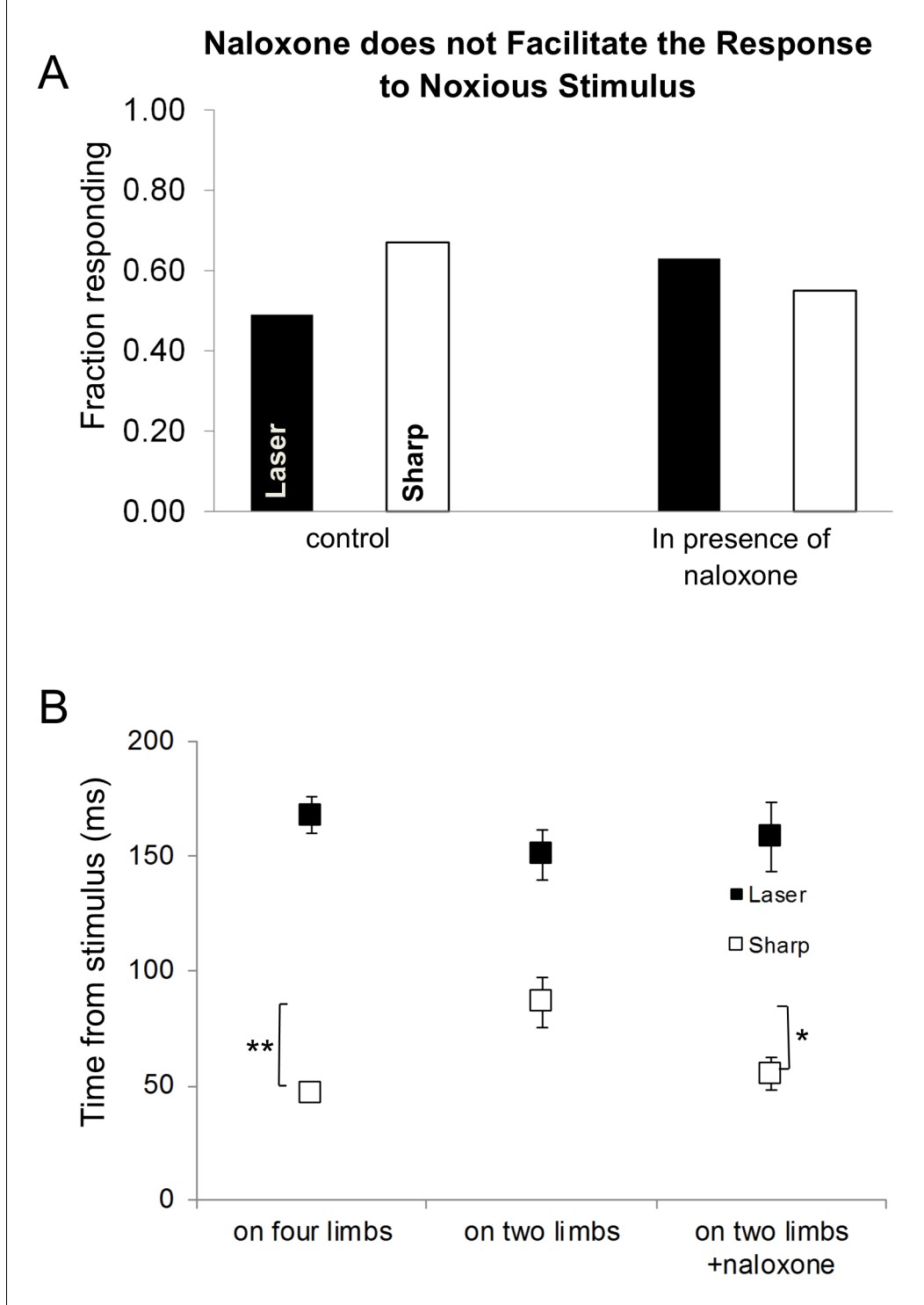

**Figure 6.** The effects of posture on the latency of the first response. (**A**) Systemic naloxone (5 mg/kg) did not enhance the response rate to noxious stimulus and did not reverse the postural inhibition of paw withdrawal when the animal was standing on its hindlimbs. In the presence of naloxone, when the rat was standing on its hindlimbs, the response rate to noxious stimuli was not statistically different from that in control conditions (laser: control vs. naloxone, p>0.5; sharp: control vs. naloxone, p>0.7; laser naloxone vs. sharp naloxone, p=1) (Bars for control conditions are adapted from *Figure 4*). (**B**) In general, the first response latency to the laser stimulus was significantly longer than that to the sharp stimuli for all postural conditions. There was no significant difference in the response latencies of animals that were standing on four vs two limbs in the presence of naloxone for laser stimulus (Laser (black rectangle): 167.5 ± 8.0 ms vs. 150.3 ± 11.0 ms vs. 158.5 ± 15.0 ms; two-way ANOVA). First response latency to the sharp stimulus was significantly longer when animals stood on their hindlimbs compared to standing on four limbs (sharp (white rectangle), four limbs– 46.9 ± 4.0 ms

*Figure 6 continued on next page*

*Figure 6 continued*

vs. on two limbs– 86.3 ± 11.0 ms; *p<0.005; two-way ANOVA). The presence of naloxone reduced significantly the average response time to noxious sharp stimuli (on two limbs +naloxone– 55.1 ± 7.0 ms; vs. on two limbs *p<0.05) but not for animals standing on four limbs (p=0.6).

# Functional and anatomical aspects of a spinal nociceptive-motor gate control circuit

## Functional considerations for a spinal nociceptive-motor gate control

The presence of the aforementioned inhibitory process prompted (a) a consideration of the necessary functional characteristics needed for multisegmental sensory-motor integration and extrapolation to adaptive significance and (b) a consideration of potential relevant neuronal populations and their characteristics. The characteristics include a capability for (i) integrating the nociceptive sensory input with multisegmental proprioceptive information, (ii) making output connections to the motor control circuits, and (iii) assessing information on the status of body stance that is instantaneously available and continuously updated as the animal moves through space. There also must be a neural registration of the postural arrangement of the limbs when the stimulus is applied, which leads to a context-dependent adjustment of the innate withdrawal strategy. The inhibition of withdrawal must be activated in a behaviorally appropriate fashion and be either maintained, terminated, or further elaborated in a contextually, posturally, and behaviorally appropriate fashion. This consideration implies the presence of inhibitory neurons and available data support the presence of both an endogenous opioid component and a GABA/glycinergic component. The second element is a capacity to resume the withdrawal sequence after the animal attains postural stability. From the bipedal stance, as a stable posture is being achieved, the spinal cord must retain the fact that a painful, possibly damaging stimulus was received during the period of transition from a bipedal stance until a stable posture conducive for limb withdrawal was achieved.

## A motor-sensory gate component

Our data showing a postural inhibition of withdrawal reflexes raises the possibility that a motor-sensory gate can regulate the behaviorally relevant multi-segmental responses to a painful stimulus. Such control could be exerted at the level of the spinal interneurons responsible for regulating the animal's upright posture. It is also possible that motoneuronal activity could contribute to such a motor-sensory gate. In the neonatal mouse, electrical stimulation of motoneuron axons can activate the central pattern generator for locomotion (*Mentis et al., 2005*; *Bonnot et al., 2009*; *O'Donovan et al., 2010*; *Pujala et al., 2016*), and this effect is blocked by the opioid agonist DAMGO (D Blivis; unpublished data). Efference copy of the motor signal, from the cortex, the red nucleus, or from spinal interneurons or even motoneurons (*Niziolek et al., 2013*; *Mason, 2017*), could be used to regulate the withdrawal reflexes. We speculate that in the extensor-dominated bipedal stance, as the nociceptive excitatory drive for limb withdrawal invades lamina IX, it encounters a 'flexor inhibited system' which must be relieved before limb flexion can occur. The inhibitory

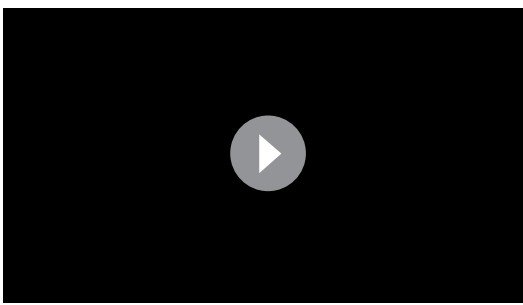

**Video 7.** Hindlimb laser stimulus when the rat is rearing up.

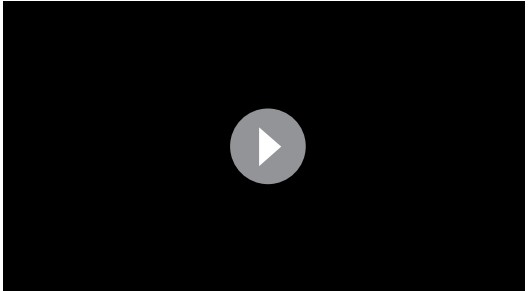

**Video 8.** Hindlimb sharp stimulus when the rat is rearing up.

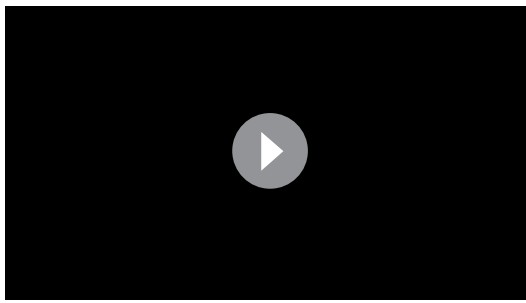

**Video 9.** Hindlimb laser stimulus when the rat is rearing up in the presence of naloxone.

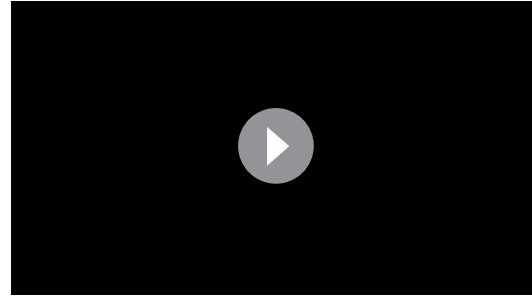

**Video 10.** Hindlimb sharp stimulus when the rat is rearing up in the presence of naloxone.

tone is relieved once the quadrupedal stance is restored. This control element is integrated with the more multi-segmental sensory-motor gating mechanisms considered below.

## Potential neural substrates of the spinal nociceptive-motor gate control circuit

A great deal of research has traced the functional responses and the primary afferent innervation of second order dorsal horn neurons with respect to a wide variety of noxious, non-noxious, hedonic, and somatosensory inputs (*Björnsdotter et al., 2010*; *Hollins, 2010*; *Stepien et al., 2010*; *Li et al., 2011*; *Arber, 2012*; *Vrontou et al., 2013*; *Levine et al., 2014*; *Pivetta et al., 2014*; *Bai et al., 2015*; *Hilde et al., 2016*). Similarly, in the ventral horn, local circuits have been extensively studied both physiologically and neuroanatomically (for review see *Jankowska, 2001*). The intricacies of the dorsal horn-ventral horn connectivity, by comparison, are only recently being addressed (*Levine et al., 2014*; *Hilde et al., 2016*). A set of interneurons in medial deep dorsal horn that meet many of the criteria delineated above appears to coordinate the execution of compound movements across multiple joints. This region was also a nexus for multimodal inputs from various muscle groups, cortico-spinal tract neurons, proprioceptive afferents, and, via synaptic relay, nociceptive afferent inputs. This heterogeneous population of neurons expresses the DNA-binding protein Satb1 and Satb2 and is in a region of the dorsal horn that has previously been shown to contain neurons regulating limb withdrawal reflexes in the rat (*Schouenborg et al., 1995*). The Satb1 neurons are inhibitory and comprise both GABA- and glycinergic neurons. They receive monosynaptic input from muscle spindle afferents and indirect inputs from nociceptors and cutaneous afferents and project to motoneurons and V1 and V2A interneurons (*Hilde et al., 2016*). When Satb2 is genetically deleted from these inhibitory interneurons their development and connectivity are severely perturbed resulting in animals with exaggerated withdrawal reflexes to noxious stimuli (*Hilde et al., 2016*). We hypothesize that these neurons may account for the postural inhibitory, non-opioid control of withdrawal reflexes and may participate in retention of the animal's balance and posture.

A second study in mouse, also using genetic labeling strategies, examined questions of sensory integratory circuits in more superficial laminae of the dorsal horn. This study anatomically and functionally defines a set of early Ret+ dorsal horn neurons mainly involved with sensory gating of nociceptive transmission to the spino-thalamic projection neurons (*Cui et al., 2016*).

Higher CNS centers can also play salient roles in controlling moment to moment monitoring and modification of spinal excitability in a coordinated fashion with the ascending sensory information. Spino-bulbo-spinal descending control mechanisms in rostral medulla have been implicated in monitoring ongoing sensory input and modifying withdrawal reflexes in the event of a painful stimulus (*Hellman and Mason, 2012*). Attentional processes also can modify pain responses to noxious stimuli and are known to have an impact when experimentally manipulated. In the present study, we did not purposely alter the animal's state of alertness or motivational status and our testing was performed an isolated room designed to minimize transient noise, air currents, temperature shifts, and people entering the room. These factors, coupled with habituation to the test platform and the testing room itself, minimized opportunities for novel environmental interferences.

Taken together, we find that changes in body posture can directly affect sensory-motor integration via a novel, motor-sensory/sensory-motor gate control mechanism operating on fast Aδ-mediated noxious input. Integration of the present findings with the above-cited studies in mouse, suggests that there are *two* gating mechanisms in the dorsal horn. One is in the more superficial laminae and gates the access of primary afferent nociceptive input to spinothalamic projection neurons, thereby modulating the relay of noxious sensory information from the cord to the brain. The other is in the deeper dorsal horn and gates the access of primary afferent nociceptive information to the motor apparatus, thereby conferring the capability for integration of the response with the ongoing and planned motor activity. These two aspects appear to be quite distinct but, as our data show, are very relevant to each other. 'Knowing' about the stimulus in terms of magnitude estimation, body location, and the context in which the painful input occurs is integral to shaping the appropriate response and the extent of muscle resources and escape strategies that may need to be employed. Our comparison of noxious to non-noxious responses is a case in point: the animal hardly attends to the blunt stimulus, whereas an abrupt painful stimulus commands abundant motoric and attentional resources.

It is important to point out that the studies identifying the Satb1/2 and early Ret[+] dorsal horn neurons were all conducted in the mouse whereas our study was performed in the rat. Nonetheless, we speculate that the nociceptive-postural gating processes investigated in this paper are fundamental to many vertebrates, be they quadrupeds or bipeds.

## Summary

Systematic evaluation of behaviors evoked by brief thermal and mechanical nociceptive stimuli reveals a coordinated interaction between noxious sensory input and the evoked multisegmental motor responses. A new inhibitory gating mechanism was also identified which we refer to as the nociceptive-motor gate control circuit. While it is possible to characterize the gate as an analgesic system, its actions are not strictly to suppress 'pain' as is commonly conceptualized, but rather to achieve a specific motor goal and, as exemplified by early orientation of the head, to perform a threat assessment. The overall function is to enable escape from a noxious stimulus while preserving balance and coordinates the elaboration of a full withdrawal response and, if necessary, additional avoidance behaviors. We show that the nociceptive-motor gate control circuit has an opioid component but also requires activation of non-opioid inhibitory elements. We enumerate the properties such a nociceptive-motor gate must possess and identify a known spinal integrator system, as fulfilling many of these requirements. It is also recognized that further elements of integration exist in the medial dorsal spinal cord such as the spino-cerebellar projections of Clark's column and projections to lateral reticular nucleus (*Pivetta et al., 2014*). The inhibitory component of the nociceptive-motor gate exerts potent control over nociceptive inputs and is activated in a context-dependent fashion. Whether the analgesic properties of the nociceptive-motor gate can be adapted for therapeutic purposes is a question open to further investigation.

# Materials and methods

## Animals

Male Sprague-Dawley rats (250–350 g; N = 9) were housed under a 12/12 hr light-dark cycle and allowed access to food and water *ad libitum*. The ambient temperature of the holding and testing rooms was 21–22°C. Procedures were performed in accordance with the National Institutes of Health (NIH) Guidelines for the Care and Use of Laboratory Animals, and approved by the institutional Animal Care and Use Committee. All efforts were made to minimize animal numbers and distress.

## Stimulus paradigms
### Thermal stimulus

Thermal stimuli (i.e. 'Laser') were generated by an infrared diode laser (LASS-10 M; Lasmed, LLC, Mountain View, CA) with an output wavelength of 980 nm and maximum power of 20 W. For calibration, laser power/energy was measured using a meter with a thermal sensor (Nova II, L30A-10 MM, Ophir Optronics). Aδ -fibers were selectively activated with a high rate of heating, using a high-energy (5000–5500 mA), brief pulse (100 ms), and a small spot size (1.6 mm Ø, nominal). As

previously reported (*Mitchell et al., 2010*), laser pulses at 7000 mA (corresponding to a power density of 6.93 W/mm$^2$) often resulted in skin damage, whereas 6000 mA (6.08 W/mm$^2$), which evokes strong behavioral withdrawal responses, never produced visible skin damage (*Mitchell et al., 2010*). At 5000 mA (5.12 W/mm$^2$) the rate of infrared laser diode heating was ~235 °C/s and no thermal damage to the skin was observed (e.g. we did not observe the presence of protein coagulation, acute redness, blistering or swelling at 24 hr post stimulation) (*Mitchell et al., 2014*).

### Pinprick stimulus

As a broad class, nociceptors are known to respond to multiple noxious stimuli whereas mechanosensitive afferents can be activated by mechanical stimuli. In the present report, we used a sharp ('Sharp') pinprick (*Magerl et al., 2001*; *Abraira and Ginty, 2013*). The tip of a sharp dissecting needle was rapidly pushed against the paw skin (*Benoliel et al., 1999*), while the animal was on an elevated wire mesh platform. The skin was dimpled but not deeply penetrated. The pin was aimed at the center of the paw to avoid failed trials. No bleeding occurred in any of the trials indicating that excessive damage is not necessary to cause a brisk withdrawal.

### Blunt stimulus

To evaluate the response to a tactile, non-nociceptive cutaneous stimulus which activates low-threshold mechanoreceptors (*Abraira and Ginty, 2013*), a thermally neutral, wooden stick with a blunt ('Blunt'), rounded tip was pushed against the glabrous skin of either the fore- or hind-paw.

### C-fiber thermal stimulation not examined

We did not evaluate a thermal stimulus that excites C-fiber afferents (*Mitchell et al., 2010*) because these all employ a slower, ramp-style of heating. Therefore, the withdrawal response is not clearly time-locked to delivery of the stimulus rendering ramp heating stimuli unsuitable for the high-speed video acquisition and analyses employed in this paper.

## Behavioral tests

Unrestrained rats were placed under plastic enclosures on an elevated glass (thermal stimulus trials) or wire mesh platform (pinprick and blunt stimulus trials). The enclosures (23 × 13 × 13 cm) were large enough for the rats to move or to stand freely (*Figure 1A*). All animals were habituated to the testing environment for 3 consecutive days prior to commencement of behavioral testing by placing them on the glass test plate under the enclosure for 10 min. On test days, stimuli were delivered following a 10-min habituation period. Thermal or mechanical stimuli were delivered with the rat's body supported either by all four limbs or only their hindlimbs for a period of at least 3–5 s before administering the stimulus (*Figure 1B*). Responses to both fore- and hindlimb stimulations were explored when the rat was resting on all four limbs. When the rat was on its hindlimbs, only its hindlimbs were stimulated. In the bipedal stance, behavioral responses to noxious stimuli were also tested after intraperitoneal injection of the opioid antagonist, naloxone hydrochloride dehydrate (5 mg/kg) (Sigma-Aldrich, USA) (*van der Kooy and Nagy, 1985*; *Hylden et al., 1991*; *McNally et al., 2000*). Naloxone was administered using a 30-g needle attached to a zero-dead volume insulin syringe (Terumo, Elkton, MD), 30 min prior to the experiment to minimize any stress-induced analgesia related to handling and the injection procedure. A control, vehicle injection was not performed. For thermal stimulation, the laser collimator, which is designed to maintain a constant beam diameter over 2 cm, was attached to a movable support and positioned below the glass, with the 1.6 mm diameter beam directed at the tip of a toe. The laser device is supplied with a red aiming beam to precisely position the infrared beam on the skin. Great care was taken to keep the glass surface dry and free of debris or excrement so that the skin could be stimulated directly by the laser.

## High-speed videography

As reported previously (*Mitchell et al., 2010*, *2014*), the response to an infrared laser stimulus that preferentially excites Aδ fibers in normal rats is characterized by rapid paw withdrawal and, with the intensities used, was frequently accompanied by orientation of the head to the stimulated paw followed by paw shaking and licking (see *Video 3*). To further quantify and characterize this sequence of events, we used a high-speed 12-bit monochrome camera (AOS Technologies AG, Switzerland) to

record limb and head movements before, during, and after the stimulus. A rectangular mirror, positioned at a 45° angle beneath the platform (*Figure 1A*), was used to direct the image of the animal onto the camera lens. Initially, images were acquired at 500 frames per second (fps) but additional experiments showed that 130–170 fps could capture the time course of the behavioral sequence with sufficient temporal resolution. A recording of 1 to 2 s encompassed the period before stimulation, delivery of the stimulus, and the animal's response and this recording was then transferred to a computer for subsequent analysis (*Figure 1C*). Recordings commenced when the rats were standing still just prior to the moment of stimulation, irrespective of their initial posture (e.g. standing on two or four legs). To quantify the timing of the limb and head movements, frames were counted from the end of the stimulus (laser, pinprick, or blunt) to the onset of the first-observable movement or muscle twitch using commercially available software (AOS Imaging studio V3.7, AOS Technologies AG, Switzerland). In addition, we determined the time of movement onset and offset for all four limbs and the head. The behavioral responses of the stimulated limb were further analyzed and divided into four types of response (withdrawal, twitch transitioning to withdrawal, twitch only, or no response). This categorization was based on the effects of the different postural states on the response.

## Data analysis

To represent the temporal relationship of the different limb movements, we constructed limb movement histograms comprised of all assays in which animals responded with a limb lifting from the platform. We set bins of 50 ms each with 0 ms corresponding to the onset of the stimulus for 'Sharp' and 'Blunt' and to the end of the 100 ms laser pulse for 'Laser' stimulus. A limb was counted as moving in all the bins from the beginning of movement until the time the movement ended or reached the set end time of 500 ms. We also calculated the mean first response latency (Mean ± SEM) for each limb and the probability of limb movement in response to the stimulus by counting the number of the trials that each limb demonstrated any type of movement divided by the total number of trials. We also broke down the type of movement (full withdrawal; twitch to withdrawal; twitch only; no movement) and its probability for the stimulated limb in response to a stimulus. Data analysis and plots were done in Matlab (Mathworks, MA, RRID:SCR_001622) and Prism6 (GraphPad Software, Inc., CA, RRID:SCR_002798). The statistics are listed in the Results section and in the figure legends. A post-hoc power analysis was done using G-Power 3.1.9.2. The two-way ANOVA analysis had a power of 0.92 for the results presented in *Figure 3* and 0.71 for the results presented in *Figure 6B*. Despite the low power, the p values for *Figure 6B* indicated statistical significance.

# Additional information

## Funding

| Funder | Grant reference number | Author |
| --- | --- | --- |
| National Institute of Neurological Disorders and Stroke | Intramural Research Program | Dvir Blivis<br>Gal Haspel<br>Michael J O'Donovan |
| National Institutes of Health | Clinical Center Intramural Research Program | Michael J Iadarola |
| National Institute of Dental and Craniofacial Research | Intramural Research Program | Philip Z Mannes<br>Michael J Iadarola |

The funders had no role in study design, data collection and interpretation, or the decision to submit the work for publication.

## Author contributions

DB, Conceptualization, Resources, Data curation, Formal analysis, Supervision, Validation, Investigation, Visualization, Methodology, Writing—original draft, Project administration, Writing—review and editing; GH, Conceptualization, Software, Formal analysis, Supervision, Investigation, Methodology, Writing—review and editing; PZM, Data curation, Formal analysis, Investigation, Visualization, Methodology, Writing—review and editing; MJO, Conceptualization, Resources, Formal analysis,

Supervision, Funding acquisition, Validation, Investigation, Methodology, Writing—original draft, Project administration, Writing—review and editing; MJI, Conceptualization, Resources, Formal analysis, Supervision, Funding acquisition, Validation, Investigation, Visualization, Methodology, Writing—original draft, Project administration, Writing—review and editing

## Author ORCIDs

Dvir Blivis, http://orcid.org/0000-0001-6203-7325
Gal Haspel, http://orcid.org/0000-0001-6701-697X
Michael J O'Donovan, http://orcid.org/0000-0003-2487-7547
Michael J Iadarola, http://orcid.org/0000-0001-7188-9810

## Ethics

Animal experimentation: This study was performed in strict accordance with the recommendations in the Guide for the Care and Use of Laboratory Animals of the National Institutes of Health. All of the animals were handled and procedures conducted according to approved animal care and use committee (ACUC) protocol (#10-555).

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
