## [Decision Letter]

Thank you for submitting your article "Identification of a novel spinal nociceptive-motor gate control circuit for Aδ pain stimuli in rats" for consideration by *eLife*. Your article has been reviewed by four peer reviewers, one of whom, Peggy Mason, is a member of our Board of Reviewing Editors, and the evaluation has been overseen and Eve Marder as the Senior Editor.

The reviewers have discussed the reviews with one another and the Reviewing Editor has drafted this decision to help you prepare a revised submission.

This is a very interesting manuscript which looks at a very simple example of reflex modulation, that of withdrawals during bipedal vs. quadrupedal stance in the rodent. The upshot is that using high speed videos, the authors show that the response to noxious stimulation is whole body, involving an orientation of the head first, followed by limb movements. They also show that the response to noxious stimulation of the hindpaw is delayed and inhibited when the rat is standing bipedally (as rats do during exploration). Of great interest is the finding that the bipedal modulation is naloxone sensitive. In sum, this manuscript greatly enriches our understanding of the organization of nocifensive responses. In particular the early engagement of the head suggest that brainstem circuits are activated as early as are spinal ones and support a spine-bulbo-spinal pathway for the spinal reactions. The naloxone-sensitivity of a standard withdrawal associated with a bipedal posture is novel and interesting.

There are two major points that we ask you to address in a revision:

The bipedal posture may be a proxy for the factor directly responsible for the modulation associated with stance. For example it could be a proxy for alertness. Please discuss this issue and acknowledge the accompanying limitations to full interpretation

The reviewers unanimously agreed that the detailed cellular circuitry claims are greatly overblown. Please reduce this type of speculation to a couple of sentences. Also change title to acknowledge this issue.

These points are fleshed out in the individual reviewers' comments below along with a list of minor points of concern.

Reviewer #1:

The authors place this work in the context of spinal circuits, particularly focusing on possible genetically identified neurons that may be involved. This reviewer finds this a stretch. Some mention of this work is fine but the experiments presented hardly provide any data that supports or contradicts the spinal neurons discussed.

The novel finding of early head involvement supports a spino-bulbo-spinal loop underlying the response to noxious stimulation. This is an old idea (Jankowska, Lundberg and others) that has been revisited more recently (see e.g. Hellman and Mason 2012). A fuller discussion of this idea would be welcome.

The finding of modulation during exploratory rearing (aka bipedal stance) could be viewed as another motor behavior during which reactions to noxious stimuli are suppressed. Other examples include micturition and eating. The authors should consider discussing their findings within this context. The naloxone sensitivity is consistent with the potential involvement of RM neurons (among a myriad of places, admittedly).

Reviewer #3:

One of the main conclusions of the authors is that the bipedal posture confers inhibitory effects on noxious stimuli-related behavior. Authors claimed that posture itself has a gating mechanism on pain. This claim is premature since it is not supported by any experimental data proving posture-induced gating at the level of the spinal cord. The alternative explanation would be that when the animal stands on its hind limbs and does not rest on its four limbs, it is because it is in a searching/exploring phase and its alertness and attention span are different than the animal resting on four feet. Different state of alertness may affect the perception of stimuli. I realize that the experiments dissecting these mechanisms are impossible to do at this stage. I therefore would suggest to the authors to leave their results as purely descriptive and leave the hypothetical interpretations in the Discussion part but not in the title, Introduction or Result parts.

Systemic pharmacological approach, or genetic, optogenetic or chemogenetic approaches in mice, would allow to draw some of the conclusion of the effects of the behavioral movement paradigm on pain perception. Authors used rats and only one drug – naloxone – which differently affected responses in bipedal or quadrupedal animals. Thus, the conclusions about the contribution of "endogenous opioid and non-opioid analgesic mechanisms" to the posture gating are preliminary.

Last, it would be interesting to see (and feasible to preform) how these movement patterns are modified in models of inflammatory and neuropathic pain, i.e. how hyperalgesia or allodynia affect these patterns.

[Editors' note: further revisions were requested prior to acceptance, as described below.]

Thank you for resubmitting your work entitled "Identification of a novel spinal nociceptive-motor gate control for Aδ pain stimuli in rats" for further consideration at *eLife*. Your revised article has been favorably evaluated by Eve Marder (Senior editor) and Peggy Mason (Reviewing editor). However, there are some remaining issues that need to be addressed before acceptance, as outlined below:

There is a strong sensory bias to this article. The authors omit self-generated movements that are not the direct result of stimulation but appear to be "spontaneous." This is nowhere more important than with respect to the central modulatory factor described here, posture. Posture is a constant that adopts different states, primarily because of self-generated movements that are not closely or even remotely tied to sensory stimuli- why does an animal choose a particular moment to rear and explore and then choose another moment to sniff at the ground? We don't know because these actions are internally generated.

The naloxone studies are too preliminary to include given that there is no saline control (I apologize for not noticing this in the first review). Rats do not receive injections in a stress-free manner and the effect of the injection procedure alone cannot be assumed to be nil. If control saline experiments are available, then another issue that should be discussed is how the rats decreased their latency when in the bipedal posture. Did they fall over?

The Reviewing Editor remains concerned regarding the use of intensity to refer to a response. The authors are urged to consider that magnitude of response is a typical term whereas intensity is typically used in reference to a stimulus. Moreover, the sentences following the use of intensity in the Introduction deal with the time domain and with the type of response and none deals with magnitude. Changing the wording only slightly would go a great length to honing this argument.

[Editors' note: further revisions were requested prior to acceptance, as described below.]

Thank you for resubmitting your work entitled "Identification of a novel spinal nociceptive-motor gate control for Aδ pain stimuli in rats" for further consideration at *eLife*. Your revised article has been favorably evaluated by Eve Marder (Senior editor), and a Reviewing editor.

The manuscript has been improved but there are some remaining issues that need to be addressed before acceptance, as outlined below:

Unfortunately the argument made regarding the lack of a saline control in the response to reviewers is 1) not in the manuscript; and 2) very indirect. Please explicitly highlight the limitations and caveats intrinsic to not having a saline control. Or omit the naloxone experiments.

---

## [Author Response]

*There are two major points that we ask you to address in a revision:*

*The bipedal posture may be a proxy for the factor directly responsible for the modulation associated with stance. For example it could be a proxy for alertness. Please discuss this issue and acknowledge the accompanying limitations to full interpretation.*

This is an interesting point to consider. There are several factors that might mitigate against interference by extraneous environmental factors.

1) The first is technical: Our testing room was isolated from transient noise, air currents, temperature shifts, and people entering the room. We also used a 10-minute habituation period on the test platform prior to testing and the animals were well acquainted with the testing room itself and the apparatus. The testing was exclusively performed by the same two people. Thus, opportunities for novel environmental interferences were minimal.

2) Second, the opportunity to be in an “altered state of alertness” or for that matter in an altered state of motoric intent (i.e., the rat might be deciding to move from its present position) is a factor whether the animal is on 4 paws or on its two hind paws. Such factors might be expected to generate a variety of patterns of response even when on 4 paws. Indeed, this is what we see. Thus, defining the exact intent or mental status of the rat at the instant of stimulation is difficult. In fact, the abrupt stimuli command the animal’s attention and they orient (within milliseconds) to the stimulus whether they were distracted or not.

(3) Third, the fact that the animal was in the “reared-up” position might be interpreted that it had shifted from a less alert to a more alert state. Therefore, the rat may be in a heightened state of awareness to any new type of somatosensory or other sensory input and theoretically could act with more alacrity rather than less. Just the opposite hypothesis as above may also apply: maybe the rat assumes a reared-up posture because it is more relaxed and comfortable in this environment leading to a decreased state of awareness and slower reactions.

Treating all of these considerations in depth may make for a lengthy addition to the paper and amount to only a rough approximation of the “thought process” of a rat at any instant in time. However, since we did not mention attention, except in the Introduction, a brief treatment of the potential impact of this parameter as discussed above has been added the Discussion.

We added the following paragraph to the Discussion.

“Spino-bulbo-spinal descending control mechanisms in rostral medulla have been implicated in monitoring ongoing sensory input and modifying withdrawal reflexes in the event of a painful stimulus (Hellman and Mason 2012). Attentional processes can also modify pain responses to noxious stimuli and are known to have an impact when experimentally manipulated. In the present study, we did not purposely alter the animal’s state of alertness or motivational status and our testing was performed an isolated room designed to minimize transient noise, air currents, temperature shifts, and people entering the room. These factors, coupled with habituation to the test platform and the testing room itself, minimized opportunities for novel environmental interferences.”

*The reviewers unanimously agreed that the detailed cellular circuitry claims are greatly overblown. Please reduce this type of speculation to a couple of sentences. Also change title to acknowledge this issue.*

We have reduced the speculation on circuits at the spinal level and added some discussion of long loop circuits as requested.

*These points are fleshed out in the individual reviewers' comments below along with a list of minor points of concern.*

*Reviewer #1:*

*The authors place this work in the context of spinal circuits, particularly focusing on possible genetically identified neurons that may be involved. This reviewer finds this a stretch. Some mention of this work is fine but the experiments presented hardly provide any data that supports or contradicts the spinal neurons discussed.*

*The novel finding of early head involvement supports a spino-bulbo-spinal loop underlying the response to noxious stimulation. This is an old idea (Jankowska, Lundberg and others) that has been revisited more recently (see e.g. Hellman and Mason 2012). A fuller discussion of this idea would be welcome.*

*The finding of modulation during exploratory rearing (aka bipedal stance) could be viewed as another motor behavior during which reactions to noxious stimuli are suppressed. Other examples include micturition and eating. The authors should consider discussing their findings within this context. The naloxone sensitivity is consistent with the potential involvement of RM neurons (among a myriad of places, admittedly).*

We appreciate this comment, which was echoed in several of the reviews. It appears we were overly detailed in our treatment of the potential circuits mediating the effect.

Several parts of the Discussion have been considerably shortened and we cite the appropriate recent publications.

In paragraph one: Changed “inhibitory circuit” to “inhibitory mechanism”.

In the same paragraph: The sentence on circuits was removed. On this page the entire circuitry discussion was shortened.

Subsection “Effects of Naloxone”: We removed the sentence containing an overly detailed consideration of potential molecular differences between DRG neurons. The citations were retained.

In subsection “Functional considerations for a spinal nociceptive-motor gate control”: The two sentences were shortened to the phrase “and (b) a consideration of potential relevant neuronal populations and their characteristics.”

In the same subsection: Two sentences were shortened into one: “This consideration implies the presence of inhibitory neurons and available data support the presence of both an endogenous opioid component and a GABA/glycinergic component”

The word circuit was changed to spinal cord.

In subsection “A motor-sensory gate component”: The potential contribution of efference copy from motoneurons was inserted. “Thus motoneuron discharge may generate an efference copy of the motor signal and this could be used to regulate the withdrawal reflex.”

The word “Potential” was added to the title:” Potential Neural substrates of the spinal nociceptive-motor gate control circuit”

Subsection “Potential Neural substrates of the spinal nociceptive-motor gate control circuit”: This section was shortened considerably but we request that some of parts be retained as they provide the reader a quick guide to a complex literature. The beginning of this section was reduced: “A set of interneurons in medial deep dorsal horn that meets many of the criteria delineated above appears to coordinate the execution of compound movements across multiple joints.” Additionally, the end of the paragraph was overly speculative and was shortened to: “We hypothesize that these neurons may account for the postural inhibitory, non-opioid control of withdrawal reflexes and may participate in retention of the animal’s balance and posture.”

The first two sentences were retained with minor modification. The last part of the paragraph again, was overly speculative and was removed. This created some space and a segue to a brief discussion of higher CNS regions and attention; these additions are discussed below.

Text removal provided room for a couple of sentences on the potential role of spino-bulbo-spinal loops and higher order influences such as attention. We added the following text: “Higher CNS centers can also play salient roles in controlling moment to moment monitoring and modification of spinal excitability in a coordinated fashion with the ascending sensory information. […] These factors, coupled with habituation to the test platform and the testing room itself, minimized opportunities for novel environmental interferences.”

*Reviewer #3:*

*One of the main conclusions of the authors is that the bipedal posture confers inhibitory effects on noxious stimuli-related behavior. Authors claimed that posture itself has a gating mechanism on pain. This claim is premature since it is not supported by any experimental data proving posture-induced gating at the level of the spinal cord. The alternative explanation would be that when the animal stands on its hind limbs and does not rest on its four limbs, it is because it is in a searching/exploring phase and its alertness and attention span are different than the animal resting on four feet. Different state of alertness may affect the perception of stimuli. I realize that the experiments dissecting these mechanisms are impossible to do at this stage. I therefore would suggest to the authors to leave their results as purely descriptive and leave the hypothetical interpretations in the Discussion part but not in the title, Introduction or Result parts.*

*Systemic pharmacological approach, or genetic, optogenetic or chemogenetic approaches in mice, would allow to draw some of the conclusion of the effects of the behavioral movement paradigm on pain perception. Authors used rats and only one drug – naloxone – which differently affected responses in bipedal or quadrupedal animals. Thus, the conclusions about the contribution of "endogenous opioid and non-opioid analgesic mechanisms" to the posture gating are preliminary.*

Last, it would be interesting to see (and feasible to preform) how these movement patterns are modified in models of inflammatory and neuropathic pain, i.e. how hyperalgesia or allodynia affect these patterns.

The title was modified and the word “circuit” was removed. The new title is: “Identification of a novel spinal nociceptive-motor gate control for Aδ pain stimuli in rats”. We acknowledge the reviewer’s concern that attentional effects might produce the inhibitory effects of posture. This was a general concern of other reviewers and we have provided a detailed response to this concern at the beginning of this document and we have now considered this point in the Discussion.

[Editors' note: further revisions were requested prior to acceptance, as described below.]

*There is a strong sensory bias to this article.*

We think that, because of the revisions we have made, the motor control elements now receive appropriate attention along the manuscript.

*The authors omit self-generated movements that are not the direct result of stimulation but appear to be "spontaneous." This is nowhere more important than with respect to the central modulatory factor described here, posture. Posture is a constant that adopts different states, primarily because of self-generated movements that are not closely or even remotely tied to sensory stimuli- why does an animal choose a particular moment to rear and explore and then choose another moment to sniff at the ground? We don't know because these actions are internally generated.*

This is an interesting point. In course of extensive testing, animals can often be observed engaging in self-generating movements. That been said, we took a great care to conduct our tests only after the animal reached, assumed, a stable quadrupedal or bipedal posture and the sensory stimulus was applied during this stable condition.

*The naloxone studies are too preliminary to include given that there is no saline control (I apologize for not noticing this in the first review). Rats do not receive injections in a stress-free manner and the effect of the injection procedure alone cannot be assumed to be nil. If control saline experiments are available, then another issue that should be discussed is how the rats decreased their latency when in the bipedal posture. Did they fall over?*

We acknowledge that injections can cause stress, and that we did not compare the results of the naloxone treated animals to saline-injected controls. However, our experimental testing was done 30min after the naloxone injection by which time stress-related analgesia will have dissipated. The action of intrathecally administered opioid peptides such as enkephalin or other proenkephalin-derived peptides lasts for ~20 min (Iadarola et al., 1986).

To clarify and acknowledge these point the following changes have been made:

“Naloxone was administered using a 30g needle attached to a zero-dead volume insulin syringe (Terumo, Elkton, MD), 30 minutes prior to the experiment to minimize any stress-induced analgesia related to handling and the injection procedure”

“Although we did not compare the results of the naloxone injections to those in saline injected control animals, we note that the experimental testing was done 30 min after the naloxone injection, which exceeds duration of analgesia evoked by many mild stressors or by intrathecally administered opioid peptides such as enkephalin or other proenkephalin-derived peptides (Iadarola et al., 1986). An opioidergic component to the postural inhibition is consistent with studies showing that withdrawal reflexes are under tonic inhibition by opioids which can be relieved by naloxone (Catley et al., 1983; Chung et al., 1983; Clarke et al., 1992; Steffens and Schomburg, 2011)”. Reference added: “Iadarola MJ, Tang J, Costa E, Yang HY. 1986. Analgesic activity and release of (MET5)enkephalin-Arg6-Gly7-Leu8 from rat spinal cord in vivo. Eur J Pharmacol. 121(1):39-48. PMID: 2420613”

*The Reviewing Editor remains concerned regarding the use of intensity to refer to a response. The authors are urged to consider that magnitude of response is a typical term whereas intensity is typically used in reference to a stimulus. Moreover, the sentences following the use of intensity in the Introduction deal with the time domain and with the type of response and none deals with magnitude. Changing the wording only slightly would go a great length to honing this argument.*

We thank the reviewer for this clarification. The following modification was made: “Time and magnitude are two, of many, domains in which human and animal behavioral responses are quantified following application of noxious stimuli and from which endpoints can be derived. However, when considering acute and chronic pain, responses in the temporal and magnitude domains occupy an extraordinarily broad range.”

[Editors' note: further revisions were requested prior to acceptance, as described below.]

*Unfortunately the argument made regarding the lack of a saline control in the response to reviewers is 1) not in the manuscript; and 2) very indirect. Please explicitly highlight the limitations and caveats intrinsic to not having a saline control. Or omit the naloxone experiments.*

As indicated by the reviewers, the argument regarding the lack of a saline control was only mentioned in the Discussion section. In order to clarify, indicate, and discuss this point appropriately along the manuscript, the following additions and changes were made:

In the Materials and method section, we added the following sentence: “A control, vehicle injection was not performed.”

The following paragraph was added to the Results section: “One limitation of these studies is that we did not use a vehicle injection as a control for the naloxone injection. Injections are anxiogenic and could influence the timing of withdrawal responses to noxious stimuli (Lapin, 1995). We note, however, that the latency of the first response to a thermal stimulus was unchanged following naloxone injection, suggesting that the effects of the injection per se did not alter the responses to all noxious stimuli. Despite this observation, we cannot eliminate the possibility that the injection might have influenced the responses to the pin prick stimulus and we consider this potential confound in more detail in the discussion.”

The following reference was added: “Lapin IP. 1995. Only controls: effect of handling, sham injection, and intraperitoneal injection of saline on behavior of mice in an elevated plus-maze. J Pharmacol Toxico Methods. 34(2):73-77. PMID: 8563035”

In the Discussion section, the explanation presented was updated to the following: “As we discussed earlier we did not use saline-injections as a control for the naloxone injections. The effect of naloxone on the latency to the first movement was only observed for the pin prick stimulus, suggesting that the injections themselves did not have non-specific analgesic effects. This may be because the experimental testing was done 30 min after the naloxone injection, which exceeds duration of analgesia evoked by many mild stressors or by intrathecally administered opioid peptides such as enkephalin or other proenkephalin-derived peptides (Iadarola et al., 1986). Consistent with this conclusion, Bryant et al., (1983) found that intrathecal saline injections did not alter the reaction latency to hot-plate test or the threshold in a paw-pressure test. While we cannot exclude a non-specific effect of the injection, our data favor the existence of an opioidergic component to the postural inhibition of the withdrawal reflex.”

The following references were added: “Bryant RM, Olley JE, Tyers MB. 1983. Antinociceptive actions of morphine and buprenorphine given intrathecally in the conscious rat. Br J Pharmacol. 78(4): 659-663. PMID: 6687818. PMCID: PMC2044748”